# Feasibility and Effectiveness of a Worksite-Weight-Loss Program for Cancer Prevention among School-District Employees with Overweight and Obesity

**DOI:** 10.3390/ijerph20010538

**Published:** 2022-12-29

**Authors:** Che Young Lee, Michael C. Robertson, Hannah Johnston, Thuan Le, Margaret Raber, Ruth Rechis, Katherine Oestman, Alise Neff, Amber Macneish, Karen M. Basen-Engquist

**Affiliations:** 1Department of Health Disparities Research, The University of Texas MD Anderson Cancer Center, Houston, TX 77030, USA; 2Department of Nutrition, Metabolism & Rehabilitation Sciences, The University of Texas Medical Branch at Galveston, Galveston, TX 77555, USA; 3Department of Behavioral Science, The University of Texas MD Anderson Cancer Center, Houston, TX 77030, USA; 4Department of Pediatrics-Nutrition, Baylor College of Medicine, Houston, TX 77030, USA; 5Be Well Comminutes, Cancer Prevention and Control Platform, The University of Texas MD Anderson Cancer Center, Houston, TX 77030, USA; 6Department of Physical Education/Health and Wellness, Pasadena Independent School District, Pasadena, TX 77502, USA

**Keywords:** weight loss, worksite intervention, community intervention, health behaviors, cancer prevention

## Abstract

The effects of Vibrant Lives, a 6-month worksite-weight-loss program, were examined in a cohort of school-district employees with overweight or obesity. The VL Basic (VLB) participants received materials and tailored text messages, the VL Plus (VLP) participants additionally received WIFI-enabled activity monitors and scales and participated in health challenges throughout the school year, and the VL Plus with Support (VLP + S) participants additionally received coaching support. The levels of program satisfaction and retention and changes in weight, physical activity (PA), and diet were compared across groups using Pearson chi-square tests, repeated-measure mixed models, and logistic regression. After the program, the VLB (*n* = 131), VLP (*n* = 87), and VLP + S (*n* = 88) groups had average weight losses of 2.5, 2.5, and 3.4 kg, respectively, and average increases in weekly PA of 40.4, 35.8, and 65.7 min, respectively. The VLP + S participants were more likely than the other participants to have clinically significant weight loss (≥3%; *p* = 0.026). Compared with the VLB participants, the VLP participants were less likely to meet the recommendations for consuming fast food (*p* = 0.022) and sugar-sweetened beverages (*p* = 0.010). The VLP and VLP + S participants reported higher program satisfaction than the VLB participants. The VL program facilitates weight loss among school-district employees with overweight and obesity by increasing their PA and healthy diet.

## 1. Introduction

Obesity is associated with several comorbidities, including cardiovascular disease and diabetes [1], and an increased risk of many types of cancer [2,3]. Physical inactivity, excessive body fat, and sedentary behavior increase the risk of cancer [4], whereas higher levels of leisure-time physical activity (PA) are associated with a lower risk of cancer, including breast, colon, and lung cancer and myeloid leukemia and myeloma [5]. Thus, among its cancer-prevention guidelines, the American Cancer Society includes recommendations on how to achieve and maintain a healthy weight, adopt a physically active lifestyle, consume plenty of vegetables and fruits, and limit the consumption of alcohol and red and processed meats [6].

Among employees with obesity, regular PA, healthy diet, and calorie reduction are effective strategies for achieving weight loss and, thus, reducing cancer risk. In the United States, the prevalence of obesity is increasing because of decreased occupational PA [7,8] and increased sedentary time [9]. In particular, teachers have a low level of occupational PA, and their rates of obesity are increasing [8]. Thus, this group could greatly benefit from programs that promote weight management and healthy behaviors. Weight management, healthy diet, and PA are important for improving teachers’ own health as well as their ability to influence their students’ health behaviors [10]. Because employed adults, including teachers and other school employees, spend an average of approximately 8 h per day at their workplaces [11], they may benefit from worksite-weight-loss programs that promote PA and healthy diets. Indeed, studies have shown that worksite-based health-promotion interventions result in weight loss, better health outcomes, and improvements in PA and eating behaviors among employees [12,13,14,15,16]. However, these programs are not always cost-effective [17], and whether they result in sustained healthy behaviors is unclear [18].

Traditional worksite-weight-loss programs that involve in-person or group-based interventions are not always scalable, which can limit their usefulness for larger groups of employees, and they can involve high cost and time burdens for both participants and providers. Since they feature few, if any, in-person activities, internet-based weight-loss interventions have smaller time and cost burdens [19]. Although internet-based weight-loss interventions have been shown to result in participant weight loss [20,21,22], the effectiveness of internet-based weight-loss programs for employees remains unclear, as such programs have been shown to have limited feasibility and efficacy, as well as low rates of participation and retention [21,23]. Therefore, a comprehensive internet-based weight-loss program designed to maximize participation and engagement is needed to help school-district employees make sustainable behavior changes that improve their weight management and cancer prevention.

The Diabetes Prevention Program (DPP) was demonstrated in a randomized controlled trial to be more effective in preventing diabetes among those at high risk than metformin or placebo [24], and has been widely used for lifestyle changes (e.g., behaviors) in health-promotion and weight-management interventions [25]. However, participant retention in the program varies by age, race/ethnicity [26], and other behavioral, psychological, and structural factors [27]. Thus, there is a need to consider adaptations and modifications to the program to identify strategies to improve retention and effectiveness [26], as well as alternative modes of implementation/program delivery to overcome barriers [28]. The Vibrant Lives program adapted the DPP program for virtual, light-touch delivery using print materials sent via email, text messages, group challenges, and weight and activity trackers, to improve retention and successful weight loss.

The aim of this study was to evaluate the effectiveness of Vibrant Lives (VL), a digital worksite-weight-loss program using the adapted DPP, in promoting weight loss, and PA, and diet among school-district employees with overweight or obesity. Participant retention in and satisfaction with the program were also assessed for the feasibility of the program.

## 2. Materials and Methods

### 2.1. Participants

This program enrolled employees of the Pasadena Independent School District, a public-school district in southeast Texas, over the 2017–2018 school year (Year 1) and the 2018–2019 school year (Year 2). Eligible participants had a BMI greater than 25 kg/m^2^ in Year 1 and a BMI greater than 27 kg/m^2^ in Year 2. The analysis of data from the program evaluation was reviewed by The University of Texas MD Anderson Cancer Center Institutional Review Board and was determined to be exempt.

### 2.2. Intervention

The VL program is a digital worksite-weight-loss program that promotes PA and healthy eating. The VL program was adapted from the Diabetes Prevention Program [29], whose recommendations for dietary intake, PA, and weight management are consistent with the cancer-prevention guidelines of the American Cancer Society [6]. The program was provided as part of the Pasadena Vibrant Community (PVC) initiative, a place-based cancer prevention program of the University of Texas MD Anderson Cancer Center’s Be Well Communities™ initiative [30]. Details of DPP in this study are described in Table 1.

The initial plan for the VL program was to offer a weight-loss program to school-district employees that included print materials from an adapted Diabetes Prevention Program (DPP, delivered by email), daily text messages that supported the DPP lessons (7–12 per week), an activity tracker (Fitbit Flex 2, San Francisco, CA, USA) and WIFI-connected scale (Fitbit Aria Smart Scale, San Francisco, CA, USA), and opportunities to participate in group challenges (Vibrant Lives Plus, VLP). In the second year, participants were also offered the opportunity to participate in a closed, secret Facebook group which provided nutrition, weight-loss, and exercise information and included weekly challenges, such as posting a recipe modified to be healthier, or a photograph of a new place to exercise. We also planned to randomly select approximately half of the schools to receive an additional support component; participants in these schools would receive an offer to participate in telephone coaching if they had not lost weight at the midpoint of the program (Vibrant Lives Plus with Support, VLP + S). However, the demand for the program was high; we had funding to provide the VLP/VLP + S program to 100 participants per year and 400 expressed interest in the first year alone. Therefore, we worked with the school-district personnel to select schools that would receive the full VLP program (with and without the additional support) to enroll approximately 120 participants each year, and the employees in the other schools were offered Vibrant Lives Basic (VLB), a low-cost version that included the emailed materials, text messages, and, in the second year, access to a Facebook group. In the first year, the school/location selection for VLP and VLP + S prioritized employees working in the administrative office to build leadership support for the program and schools that were implementing the Coordinated Approach to Child Health (CATCH) program [31]. Selection of the schools/locations for the second year prioritized schools/locations not selected in the first year (Table 2). Enrolled employees participated in one of the three versions of the VL program. Participants in the Vibrant Lives Basic (VLB) group received emails with the DPP materials and text messages (7–12 per week) with tips for behavior change that were coordinated with the DPP lessons.

### 2.3. Measures

We used a survey to collect participants’ demographic data, including age, race, sex, education, and marital status. Participants reported their self-measured weights at baseline and follow-up. Participants in the VLP and VLS groups self-measured their weights on their WIFI-enabled scales.

Participants reported their PA and dietary-intake behaviors at baseline and follow-up using the survey items from the Health Information National Trends Survey [32] and Health of Houston Survey [33]. Self-reports of PA included the number of minutes of moderate-to-vigorous aerobic activity per day and the number of times in which strength exercise was performed per week. Self-reports of dietary intake included the numbers of times red meat, fast food, and sugar-sweetened beverages were consumed per week, the number of cups of fruit and vegetables that was consumed per day, and the number of days breakfast was consumed during a week.

We scored participants’ responses on the reports according to whether they met the recommendations of at least 150 min of moderate-to-vigorous PA per week and at least two sessions of strength exercise per week (meet, score = 1; did not meet, score = 0); how often they consumed red meat (≤2 times/day, score = 1; ≥3 times/day, score = 0), fast food (≤2 times/week, score = 1; ≥3 times/week, score = 0), and sugar-sweetened beverages (never in the week, score = 1; at least once in the week, score = 0); their daily consumption of fruit (≥1 cup/day, score = 1; <1 cup/day, score = 0) and vegetables (≥3 cups/day, score = 1; <3 times/day, score = 0); and how often they ate breakfast (every day in the past week, score = 1; 0–6 days in the week, score = 0).

Participants in the VLP and VLP + S groups used their Fitbits to measure their physical activity. After excluding non-valid days (those with <1500 steps or <10 h of valid wear) and non-valid weeks (those with <4 days of valid wear), we summed the number of daily steps and “very active”, “fairly active”, and “lightly active” minutes. “Active minutes” represented the combination of the number of Fitbit-derived “fairly active” and “very active” minutes per day. Detailed information about Fitbit-data use in the VL program is provided elsewhere [34].

The program-retention rate was the percentage of participants who completed both the baseline and follow-up assessments and a program-satisfaction survey. Program satisfaction was assessed with a Likert-scale survey (1 = strongly disagree; 2 = disagree; 3 = neutral; 4 = agree; 5 = strongly agree) with the following prompts: “Being in the program has motivated me to increase my physical activity”; “Being in the program has motivated me to adopt a healthy diet”; “Being in the program has motivated me to lose weight”; “This program was effective for me”; and “I would recommend this program to a friend or family member”.

### 2.4. Statistical Analyses

We used descriptive statistics to summarize the baseline characteristics of the participants. One-way analysis of variance and the Pearson’s chi-square test were used to detect differences in continuous variables (e.g., age, BMI) and categorical variables (e.g., race, sex, education, BMI category), respectively, between the participants who completed the program, “completers”, and those who did not complete the program, “non-completers”, among the three groups. Changes in weight, PA, and dietary intake between baseline and follow-up among the VLB, VLP, and VLP + S groups were analyzed using multi-level repeated-measure linear mixed models and logistic-regression models; in these analyses, we clustered participants by school and controlled for the study years (e.g., Year 1 and Year 2) and race/ethnicity as covariates. Differences in program retention and satisfaction among the three groups were assessed using the Pearson’s chi-square test. Fitbit-derived daily step counts and active minutes were converted to weekly average, which were reported using linear regressions for each of the 26 weeks of the program and compared between VLP and VLP + S groups using repeated-measure mixed models. All statistical analyses were performed with the STATA 15.1 statistical software program (StataCorp LP, College Station, Texas). Results were reported using an unstandardized beta (β) or odds ratio (OR) with a standard error (SE) and *p*-value (*p* < 0.05).

## 3. Results

Of the 543 participants initially enrolled in the VL program, 306 (131, 87, and 88 in the VLB, VLP, and VLP + S groups, respectively) completed the program and follow-up assessments and were included in the analyses. The percentage of program completion (program retention) by the groups is shown in Figure 1. The retention rate in the VLB group (43%) was significantly lower than that in the VLP group (73%; χ^2^ = 30.85, *p* < 0.001) and that in the VLP + S group (73%; χ^2^ = 31.51, *p* < 0.001).

The baseline characteristics of the completers and non-completers are provided in the Table 3; no significant differences between the groups were detected.

The characteristics of the 306 completers by intervention arm are provided in Table 4. Compared with the VLB and VLP + S groups, the VLP arm had a significantly higher proportion of participants who were Hispanic White or non-Hispanic White (χ^2^ = 20.48, *p* = 0.025); no other significant differences among the groups were detected.

The changes in weight, PA, and dietary intake between baseline and follow-up are shown in Figure 2. The VLB, VLP, and VLP + S groups all had significant weight losses (mean, 2.5, 2.5, and 3.4 kg, respectively; β = –2.34; SE = 0.45; *p* < 0.001) and increases in weekly moderate-to-vigorous PA (mean, 40.4, 35.8, and 65.7 min, respectively; β = 40.53; SE = 8.00; *p* < 0.001), but these variables did not differ significantly among the groups. The participants in the VLP + S were more likely to have clinically significant weight loss (≥3%) than the other participants (OR: 1.47; SE = 0.28; *p* = 0.045). All the groups had higher rates of moderate-to-vigorous PA (≥150 min/week of aerobic exercise; OR = 2.31, SE = 0.54, *p* < 0.001) and strength exercise (≥2 times/week; OR = 2.11, SE = 0.45, *p* < 0.001) at the end of the program, but these rates did not differ significantly among the groups. In addition, all the groups had significantly higher rates of consuming red meat two or fewer times per week (OR = 2.07; SE = 0.39; *p* < 0.001), consuming fast food two or fewer times per week (OR = 4.22; SE = 0.92; *p* < 0.001), and not consuming sugar-sweetened beverages in the past week (OR = 1.65, SE = 0.40, *p* = 0.037). They also had higher rates of consuming at least one cup of fruit per day (OR = 1.90; SE = 0.36; *p* = 0.001) and of consuming at least three cups of vegetables per day (OR = 2.00; SE = 0.44; *p* = 0.002). Compared with the VLB group, at the end of the program, the VLP group had lower rates of consuming fast food fewer than two times per week (OR = 0.48. SE = 0.15, *p* = 0.022), and not consuming sugar-sweetened beverages in the past week (OR = 0.46, SE = 0.14, *p* = 0.010).

The weekly average numbers of daily active minutes and steps for the VLP and VLP + S groups (*n* = 175) are given in Figure 3. Overall, the average number of daily active minutes was 20 min (SD = 23.1; VLP = 17 min ± 18.9; VLP + S = 23 min ± 25.7), and the average number of daily steps was 8030 (SD = 2939.9; VLP = 7807 steps ± 2794.1; VLP + S = 8212 steps ± 3042.2). There were significant differences in daily active minutes (β = 6.72; SE = 3.06; *p* = 0.028) and daily steps (β = 969.40; SE = 390.44; *p* = 0.013) between the VLP and VLP + S groups. The linear rate of daily active minutes was significant for the VLP group (β = 0.15; SE = 0.06; *p* = 0.007), but not for the VLP + S group (β = 0.02; SE = 0.07; *p* = 0.790). The linear rate of the daily number of steps was not significant for the VLP group (β = 15.57; SE = 8.17; *p* = 0.057) or for the VLP + S group (β = −3.38; SE = 8.03; *p* = 0.673).

The program-satisfaction rates of the VLP and VLP + S groups were significantly higher than that of the VLB group for each statement (Table 5).

## 4. Discussion

The VL program resulted in weight loss as well as positive changes in PA and diet in an ethnically diverse employee population, regardless of the version of the intervention they received. These findings are in line with those of other studies of internet-based worksite interventions [16,35,36,37], and build on them, as this study is one of the few to focus on a school setting in a racially and ethnically diverse community, in which 71% of the population are Hispanic and 18.2% live below the poverty line (compared to 12.8% nationally) [38].

The present study evaluated a DPP behavior-change intervention adapted to be remotely delivered using digital strategies to encourage weight loss and healthier behaviors among school-district employees with overweight and obesity. The DPP was translated into practice in the workplace and showed its feasibility and effectiveness for weight loss. Two previous studies demonstrated that adaptations and modifications of DPP improved feasibility and effectiveness: minimizing program intensity and costs and varying delivery formats, such as small face-to-face groups, online, or telephone coaching improved retention rates [39,40]. However, the results of the adaptations are mixed: face-to-face group-based intervention showed higher retention but no group differences in weight loss [39], while telephone coaching showed more weight loss than the use of a face-to-face small group [40]. In our study, the digital delivery of DPP content resulted in weight loss, increases in PA, healthier diets, and high retention. Weight loss in the VL interventions resulted in greater or comparable weight loss (−3.4 to −2.5 kg) to these two studies (Ing et al.: −0.48 to −0.07 kg [39]; Wilson et al.: −2.26 to −1.22 kg [40]).

Internet-based worksite PA interventions have shown higher attrition rates and lower participation rates [21,23]. However, Tate et al. showed that, compared with participants who were randomized to a weight-loss program alone, participants who additionally received counseling through emails, messages, virtual discussion boards, and other feedback had more weight loss and higher participation rates [41]. In the present study, participants with access to more interactive program components did not differ significantly in weight loss or changes in physical activity from those who completed the VLB program. However, the retention rate in the VLB group was dramatically different, with more than half of the participants (57%) dropping out before the end of the program. Other studies have shown that self-monitoring strategies that employ wearable fitness-tracking devices may help participants achieve their goals and motivate them to lose weight and become physically active [42,43]. In accordance with their higher retention rates, the VLP and VLP + S groups also had higher satisfaction rates than the VLB group did. Thus, the interactive components of the program, such as the activity monitor, connected scale, and challenges based on the data these devices provided, appeared to increase program retention and engagement.

At the end of the VL program, the participants in the VLP + S group had greater weight loss and a greater increase in the number of minutes of moderate-to-vigorous PA than the participants in the VLB or VLP groups, but these differences were not significant. In addition, compared with the VLP group, the VLB group had higher percentages of participants who consumed fast food two or fewer times per week and who had not drunk a sugar-sweetened beverage in the past week. These unexpected results may have been due to the high level of attrition in the VLB group, whose retention rate (43%) was significantly lower than those of the VLP and VLP + S groups (both 73%). Some of the outcomes of the VLB group may have been better than those of the VLP group because the 43% of VLB participants who completed the program may have been more motivated than the other subsets of participants, who received the two other versions of the program.

During the program, the participants in the VLP and VLP + S groups logged more than 8000 steps per day on average, which is in line with a reasonable threshold of free-living PA (7000–8000 steps/day) and with current PA guidelines [44]. The Fitbit-derived numbers of daily steps and active minutes varied weekly; in particular, the average numbers of steps and active minutes were lower from weeks 9 to 11 and from weeks 21 to 22. These periods corresponded to a school holiday at the end of December (Winter Break) and another in mid-March (Spring Break), respectively. Previous studies of worksite interventions that used pedometers also showed that employees’ step counts decreased during the winter [45], which may have been due to employees taking fewer steps outside work during the holidays [46]. Given these patterns of PA, future participants in the VL and similar programs should be encouraged to remain physically active and engage in extra activity during holidays such as Spring Break and Winter Break. The VLP and VLP + S groups participated in a “no gain” challenge over the holidays, in which they received prizes if they did not gain weight between Thanksgiving and New Year’s, but in future, efforts including physical-activity maintenance may help to boost this effect.

Some limitations should be considered when interpreting the results of our study. First, the participants’ self-reported PA did not capture increases in light PA or reductions in sedentary behavior, which also burns calories and can aid in weight loss. Second, the measurement of diet was limited by the need to reduce the assessment burden in this worksite-delivered program. The dichotomous variables of dietary intake, which indicated meeting healthy-eating recommendations, did not enable a direct association between diet changes and weight loss among the participants because the items do not capture all intake and do not allow the calculation of energy intake. Third, because all the participants were school-district employees from a specific geographic region (southeast Houston, in Texas) and most were women, it would be difficult to generalize our findings to other populations. Finally, because the study lacked a control arm, we cannot be certain that the effects we observed were caused by the intervention or by other programs or secular trends. However, given that adults in the US typically gain approximately one pound per year [46,47], it seems unlikely that the weight loss would have occurred without intervention. Nevertheless, our study shows that, given the relatively high rates of program retention and satisfaction, particularly in the VLP and VLP + S versions, which had more interactive components, the VL program was generally well received. In addition, its findings provide new evidence to help investigators and practitioners consider the use of effective and feasible worksite weight-loss and behavior-change-intervention programs.

## 5. Conclusions

The VL program is a promising approach to facilitating weight loss among school-district employees with overweight and obesity by increasing their PA and healthy eating behaviors. This scalable, light-touch intervention may help employees improve their behaviors, achieve their weight-loss goals, and, consequently reduce their cancer risk. While all the versions of the program had similar effects on weight loss and physical activity among those who completed it, the inclusion of more interactive components (e.g., activity monitors, scales, and challenges) may increase retention.

## Figures and Tables

**Figure 1 ijerph-20-00538-f001:**
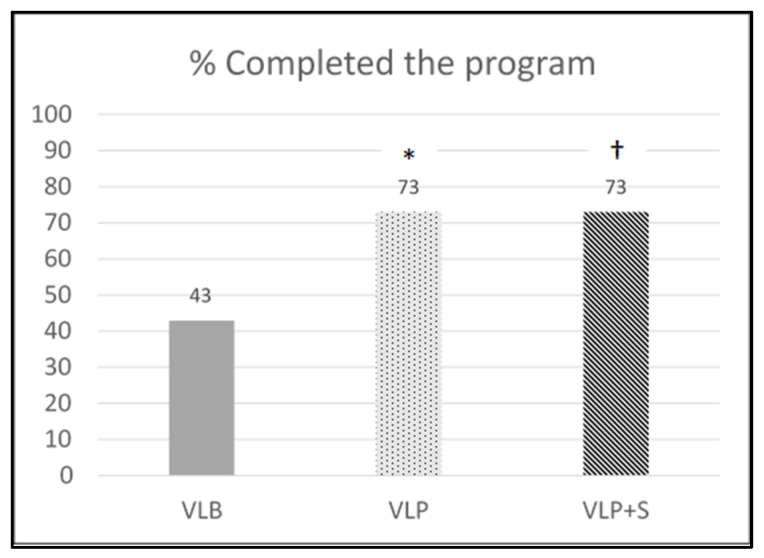
Retention rate by intervention group. VLB = Vibrant Lives Basic; VLP = Vibrant Lives Plus; VLP + S = Vibrant Lives Plus with Support; * significant differences between VLB and VLP groups; † significant differences between VLB and VLP + S groups.

**Figure 2 ijerph-20-00538-f002:**
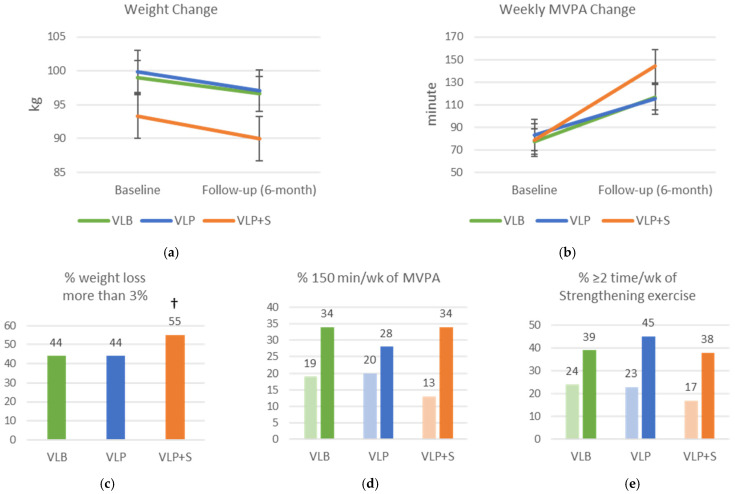
Changes in weight, physical activity, and dietary intake among intervention groups between baseline and follow-up (6-month). (**a**) Mean weight change; (**b**) mean weekly moderate-to-vigorous physical activity (MVPA) change; (**c**) percentage who lost more than 3% of their weight; (**d**) percentage who performed 150 min/week of MVPA; (**e**) percentage who performed two or more sessions per week of strengthening exercise; (**f**) percentage who consumed red meat two or fewer times in the past week; (**g**) percentage who consumed one or more cups of fruit per day; (**h**) percentage who consumed three or more cups of vegetables per day; (**i**) percentage who consumed fast food two or fewer times in the past week; (**j**) percentage who ate breakfast every day in the past week; (**k**) percentage who drank no sweetened beverages in the past week; VLB = Vibrant Lives Basic; VLP = Vibrant Lives Plus; VLP + S = Vibrant Lives Plus with Support; (**a**,**b**) repeated-measure mixed models; (**c**) logistic regression; (**d**–**k**) repeated measure logistic regressions; except for breakfast consumption, all had significant time differences (*p* < 0.05). † Significant group differences compared to VLB arm (*p* < 0.05).

**Figure 3 ijerph-20-00538-f003:**
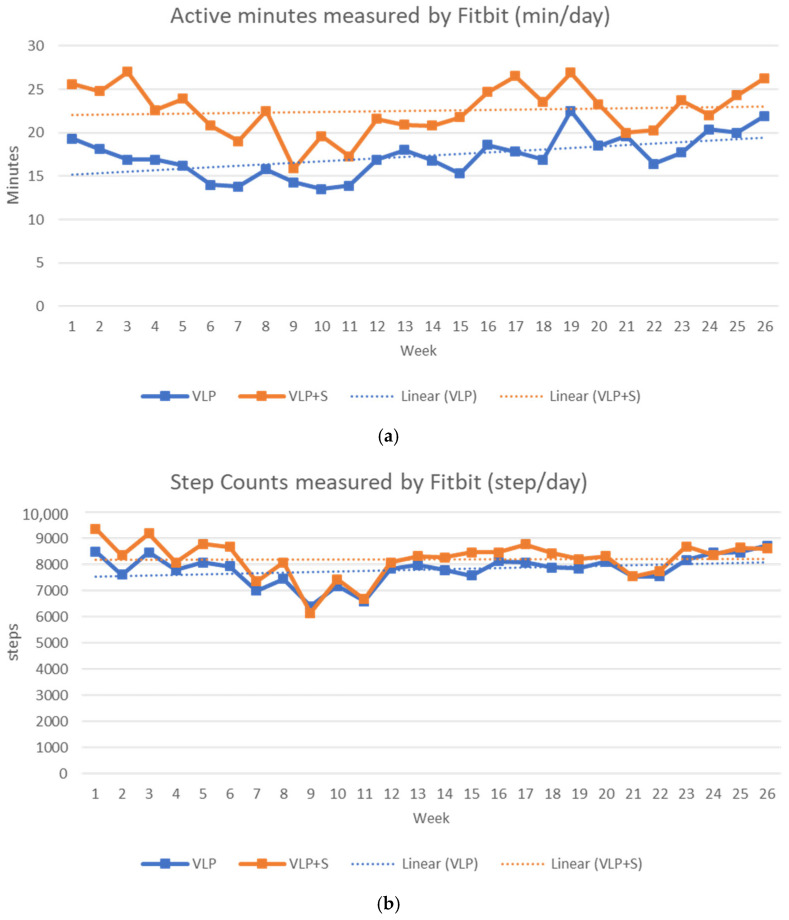
Number of Fitbit-derived active minutes (”fairly active” plus “very active” minutes) and steps per day by intervention group. (**a**) Active minutes measured by Fitbit (min/day); (**b**) step counts measured by Fitbit (step/day); VLP = Vibrant Lives Plus; VLP + S = Vibrant Lives Plus with Support.

**Table 1 ijerph-20-00538-t001:** Adapted Diabetes Prevention Program (DPP) details.

Lessons ^1^	Program Keys and Details
1: Welcome to Vibrant Lives!	Program overview and goals
2: Move those Muscles	Activity goals (150 min/week or 10,000 steps/day)
3: Be A Fat and Calorie Detective	Reducing fat consumption and calories
4: Being Active–A Way of Life	Be active and reduce sedentary time
5: Three ways to Eat Less Fat and Fewer Calories	Eat foods high in fat less often, in smaller amounts, and eat lower-fat and lower-calorie foods
6: Healthy Eating	The New American Plate (2/3 of vegetables, fruits, whole grains or beans and 1/3 of animal protein)
7: Tip the Calorie Balance	Tracking calories-in (eating) and -out (activity) for weight loss
8: Take Charge of What’s Around You	Understanding cues and changing food consumption and activity habits
9: Problem Solving	Describing a problem, listing options, choosing one to try, making a positive action plan, and solving it
10: Four Keys to Healthy Eating Out	Making choices: plan ahead, ask for what you want, take charge of what is around you, and choose food carefully
11: Talk Back to Negative Thoughts	Practicing talking back negative thoughts and talking back with positive thoughts
12: Slippery Slope of Lifestyle Change	Understanding slip and revisiting your calorie and fat consumption and activity goals
13: Jump Start Your Activity Plan	Considering frequency, intensity, time, and type of activity to avoid boredom and improve fitness
14: Make Social Cues Work for You	Understanding social cues and dealing with problem social cues
15: You Can Manage Stress	Preventing possible stress and solving the stress
16: Ways to Stay Motivated	Reviewing progress, revisiting your goals, and planning to stay motivated

^1^ All groups were provided with lesson handouts via email, and VLP and VLP + S were additionally provided supplementary handouts for the special weeks; Lessons 1–8 were provided weekly, and Lessons 9–16 were provided bi-weekly.

**Table 2 ijerph-20-00538-t002:** Vibrant Lives program details.

	2017–2018 (Year 1)	2018–2019 (Year 2)
Program Start and End Date	13 November 2017–30 April 2018	29 October 2018–30 April 2019
**Program Groups**		
Vibrant Lives Basic (VLB)	x	x
Vibrant Lives Plus (VLP)	x	x
Vibrant Lives Plus with Support (VLP + S)	x	x
**Fitbit Devices Distributed**		
Fitbit Flex	x	x
Fitbit Aria 1	x	
Fitbit Aria 2	x	x
**Program Materials**		
DPP Lessons ^1^	x	x
Text Messages ^2^	x	x
**Support Coaching ^3^**		
Support Calls from Dietitian	x	x
Support Emails from Dietitian		x
Support Calls from Physiologist		x
Support Emails from Physiologist		x
**Program Challenges**		
Self-Monitoring		x
No Gain	x	x
Steps to Heart Health	x	x
Spring into Action	x	x
Commit to be Fit	x	x
**Private Facebook Groups**		
Weight-Loss Group ^4^		x
Maintenance Group ^5^	x	x

^1^ DPP Lessons updated using UT MD Anderson Cancer Center Creative Services. The updated DPP was implemented in Year 2. ^2^ Text Messages updated based on feedback (fewer messages, more images, and links to resources). ^3^ Weight-Loss Group is defined as Vibrant Lives private Facebook Group during the program. Separate groups were created based on group assignment. ^4^ Maintenance Group was for Vibrant Lives participants who completed the VL program. This was active after program end date. ^5^ Support Coaching in Year 1 was staff dietitian. Participants were monitored on their weight change and offered support coaching at week 12. In Year 2, an exercise physiologist was added. In addition, coaching via phone call or email was offered. Coaching was offered earlier, beginning in Week 6. x = included; blank = not included.

**Table 3 ijerph-20-00538-t003:** Baseline characteristics of Vibrant Lives completers and drop-outs.

Characteristic	Completers (*n* = 306)	Non-Completers (*n* = 237)	*p*-Value
Mean age, years (SD)	42.6 (10.3)	42.5 (9.9)	0.915
Race, *n* (%)			0.687
Hispanic White	119 (39)	87 (36)	
Non-Hispanic White	133 (43)	115 (49)	
Hispanic Black	3 (1)	1 (1)	
Non-Hispanic Black	34 (11)	23 (10)	
Asian	8 (3)	3 (1)	
Other	9 (3)	8 (3)	
Female sex, *n* (%)	278 (91)	212 (90)	0.701
Education, *n* (%)			0.453
HS diploma/GED or less	31 (10)	20 (8)	
Technical/vocational degree	7 (2)	4 (2)	
Some college	54 (18)	35 (15)	
Bachelor’s degree	105 (34)	101 (43)	
Master’s degree	99 (33)	69 (29)	
Doctoral degree	10 (3)	8 (3)	
Marital status, *n* (%)			0.341
Single	62 (20)	42 (18)	
Married/cohabiting	203 (67)	153 (65)	
Divorced/separated	41 (13)	39 (16)	
Widowed	0 (0)	3 (1)	
BMI category, *n* (%)			0.717
Overweight	67 (22)	55 (23)	
Obese	239 (78)	182 (77)	
Mean BMI, kg/m^2^ (SD)	35.7 (7.0)	36.8 (7.7)	0.086

BMI = body-mass index; GED = general educational development; HS = high school; SD = standard deviation.

**Table 4 ijerph-20-00538-t004:** Baseline characteristics of participants among intervention groups (*n* = 306).

Characteristic	VLB (*n* = 131)	VLP (*n* = 87)	VLP + S (*n* = 88)	*p*-Value
Mean age, years (SD)	41.9 (10.1)	41.9 (9.3)	44.3 (11.2)	0.178
Race, *n* (%)				0.025
Hispanic White	53 (41)	40 (46)	26 (30)	
Non-Hispanic White	47 (36)	38 (44)	48 (54)	
Hispanic Black	2 (1)	0 (0)	1 (1)	
Non-Hispanic Black	21 (16)	7 (8)	6 (7)	
Asian	2 (1)	1 (1)	5 (6)	
Other	6 (5)	1 (1)	2 (2)	
Female sex, *n* (%)	119 (91)	79 (91)	80 (91)	1.000
Education, *n* (%)				0.463
HS diploma/GED or less	8 (6)	13 (15)	10 (11)	
Technical/vocational degree	2 (1)	1 (1)	4 (5)	
Some college	26 (20)	15 (17)	13 (15)	
Bachelor’s degree	48 (37)	25 (29)	32 (36)	
Master’s degree	43 (33)	29 (33)	27 (31)	
Doctoral degree	4 (3)	4 (5)	2 (2)	
Marital status, *n* (%)				0.689
Single	26 (20)	16 (18)	20 (23)	
Married/cohabiting	85 (65)	62 (71)	56 (63)	
Divorced/separated	20 (15)	9 (11)	12 (14)	
Widowed	0 (0)	0 (0)	0 (0)	
BMI category, *n* (%)				0.751
Overweight	30 (23)	16 (18)	21 (24)	
Obese	101 (77)	71 (82)	67 (76)	
Mean BMI, kg/m^2^ (SD)	35.7 (7.3)	36.2 (6.4)	35.4 (7.2)	0.637

VLB = Vibrant Lives Basic; VLP = Vibrant Lives Plus; VLP + S = Vibrant Lives Plus with Support; BMI = body-mass index; GED = general educational development; HS = high school; SD = standard deviation.

**Table 5 ijerph-20-00538-t005:** Vibrant Lives program satisfaction by intervention group.

Statement ^1^	VLB (*n* = 131)	VLP (*n* = 87)	VLP + S (*n* = 88)	*p*-Value
Being in the program has motivated me to increase my physical activity.	77 (59)	65 (75)	74 (84)	<0.001
Being in the program has motivated me to adopt a healthy diet.	74 (57)	64 (74)	70 (80)	0.001
Being in the program has motivated me to lose weight.	78 (60)	70 (80)	71 (81)	<0.001
This program was effective for me.	54 (42)	53 (61)	56 (64)	0.001
I would recommend this program to a friend or family member.	70 (54)	69 (79)	75 (85)	<0.001

VLB = Vibrant Lives Basic; VLP = Vibrant Lives Plus; VLP + S = Vibrant Lives Plus with Support. ^1^ Data are the number of participants (%) who responded “agree” or “strongly agree” to the prompts on the program-satisfaction survey.

## Data Availability

The datasets used and analyzed during this study are available from MD Anderson Cancer Center and Karen M. Basen-Engquist upon reasonable request.

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
