# Peer review of "Feasibility and Effectiveness of a Worksite-Weight-Loss Program for Cancer Prevention among School-District Employees with Overweight and Obesity"

_ijerph, 2022, doi:10.3390/ijerph20010538_

Round 1
Reviewer 1 Report
Introduction:
I recommend that the DPP be introduced and described early-on, in the introduction. Basing your program off of the DPP is a strength and could be better highlighted and described. An important point to include, with references, is that the DPP is an effective program among program completers, however program completion can be a challenge. This provides an excellent rationale for adapting/modifying the program to increase retention and engagement.
A sample of the DPP program lessons could be added as a table so that the audience gets a better sense of the intervention content.
Page 2, line 60: There seems to be an error in this sentence: "... whether the results in sustained behaviors is unclear."
Page 5, line 198: Revision needed. Completers and non-completers.
Page 10, lines 301-304: Revision needed in this sentence: "...may have been more motivated than other participants subset of the participants than those..." ??
Pages 10-11, lines 314-315: Revision needed in this sentence.
Page 11, line 327: Pound instead of point?
Discussion: Could the authors compare and contrast to published research describing the implementation of adaptations of the DPP in workplace wellness initiatives?
Author Response
Reviewer 1: Comments and Suggestions for Authors
Introduction:
I recommend that the DPP be introduced and described early-on, in the introduction. Basing your program off of the DPP is a strength and could be better highlighted and described. An important point to include, with references, is that the DPP is an effective program among program completers, however program completion can be a challenge. This provides an excellent rationale for adapting/modifying the program to increase retention and engagement.
A sample of the DPP program lessons could be added as a table so that the audience gets a better sense of the intervention content.
Thanks for your great points and suggestions. We added a paragraph for brief introduction of DPP in the Introduction section and a table (Table 1) for DPP program details that were used in the VL program in the Method section.
Page 2, line 60: There seems to be an error in this sentence: "... whether the results in sustained behaviors is unclear."
The sentence was revised as “However, such programs are not always cost-effective [17], and whether they result in sustained healthy behaviors is unclear [18].”
Page 5, line 198: Revision needed. Completers and non-completers.
The sentence was revised as you pointed out.
Page 10, lines 301-304: Revision needed in this sentence: "...may have been more motivated than other participants subset of the participants than those..."
The ‘a’ was removed from the sentence as you suggested.
Pages 10-11, lines 314-315: Revision needed in this sentence.
The sentence was revised as “Given these patterns of PA, future participants in the VL and similar programs should be encouraged to remain physically active and engage in extra activity during holidays such as Spring Break and Winter Break.”
Page 11, line 327: Pound instead of point?
The point was changed to pound.
Discussion: Could the authors compare and contrast to published research describing the implementation of adaptations of the DPP in workplace wellness initiatives?
We added a paragraph for comparisons of previous studies that used DPP in the Discussion.
Reviewer 2 Report
This is an interesting study that evaluates physical activity and behavioral changes as a means of weight loss in overweight people. The study has an adequate design that analyzes interesting variables. Statistical analysis is adequate for the study design.
My main consideration is that there was no quantitative control on the individuals' food consumption. Although the frequency of consumption of sweetened drinks and consumption of fast foods was evaluated, the categorical evaluation does not allow inferring whether weight loss was also associated with food consumption. It is an important limitation that should be mentioned in the discussion and limitations of the study. I understand that it was not the research objective of this study, but weight loss in overweight people requires a multidisciplinary assessment to understand the whole context.
Author Response
Reviewer 2: Comments and Suggestions for Authors
This is an interesting study that evaluates physical activity and behavioral changes as a means of weight loss in overweight people. The study has an adequate design that analyzes interesting variables. Statistical analysis is adequate for the study design.
My main consideration is that there was no quantitative control on the individuals' food consumption. Although the frequency of consumption of sweetened drinks and consumption of fast foods was evaluated, the categorical evaluation does not allow inferring whether weight loss was also associated with food consumption. It is an important limitation that should be mentioned in the discussion and limitations of the study. I understand that it was not the research objective of this study, but weight loss in overweight people requires a multidisciplinary assessment to understand the whole context.
Thanks for bringing this important point up. We added it in the limitation of the study.